# Supplementing dairy feed by dicalcium phosphate and effect on dry matter intake, digestibility, milk composition, and blood mineral balances in crossbred dairy cows

**Wondewsen Bekele Wondater**[1,2]*, **Tilaye Demissie Ayanie**[2]

**1** Animal Sciences Department, College of Agriculture and Natural Resources, Dilla University, Dilla, Ethiopia, **2** Animal Production Studies, College of Veterinary Medicine and Agriculture, Addis Ababa University, Addis Ababa, Ethiopia

* wondewsen19@gmail.com

**Data Availability Statement:** The authors declare that all data supporting the findings of this study

## Abstract

The present study was conducted to assess if Dicalcium Phosphate 18% (DCP) supplementation in dairy feed would affect dry matter intake, milk composition, blood mineral balance and milk production in lactating crossbred Dairy Cows). A 4 × 4 double change-over design set as Latin Squares was used to do feeding and digestibility trials. DCP was supplemented at different levels 0; 0.3; 0.6; 0.9% DM/day of DCP for which the experimental groups was respectively designated as T0, T1, T2, T3, The effect of DCP supplementation on parameters of interest was evaluated using blood serum, milk, feed and feed refusals samples. The present study revealed that supplementation of DCP with the concentrate mix improved mean dry matter intake (11.46; 12.81; 14.59, and 15.68 kg DM/d) for T0, T1, T2, T3, respectively, and digestibility of nutrients in lactating dairy cows compared to the controls. Likewise, DCP supplementation improved milk production and composition except solids not fat (SNF) composition. Serum calcium (Ca) composition was significantly in supplemented group ($p < 0.05$). DCP supplementation significantly improved ($p < 0.05$) total Ca intake, Ca in milk, Ca in feces, total Ca excretion. Optimum dairy cow production and productivity was achieved at 0.6% DM/day of DCP supplementation. However, the current results need to be confirmed using large number of lactating dairy animals.

## Introduction

In Ethiopia, over 90% of milk is produced by smallholder producers [1]. However, dairy production is much limited due to factors like feed shortage, mineral deficiencies, mineral imbalances, seasonal inadequacy of the quantity and quality of available feed resources and some other management issues [2]. Nutrients like minerals, carbohydrates, proteins, fat, vitamins, and water are equally imperative as one or more of these can hamper the productivity of dairy cows when requirements are not fulfilled [3]. Minerals may constitute a small fraction of the total ration, but these micronutrients perform a vital role in the biochemical process in the

are available within the article and its supplementary information files.

**Funding:** The authors received funding from Adis Abeba university, AGP-II (Agricultural Growth Program) through Debreziet Agricultural institute and Dilla university. Adis Abeba University support the author through the thematic project of Dr Tilaye Demissie who is included in publication(Advisor). However, the AGP-II (Agricultural Growth Program) through Debrezeit Agricultural research allowed for the author dairy cow farm to perform my feeding trial simply because the Debrezeit Agricultural research need the output of the research and I did that as recommendation.

**Competing interests:** The authors have declared that no competing interests exist.

body. However, in many smallholder and commercial dairy production systems of Ethiopia, the feed being provided for dairy is believed to be imbalanced and, in many instances, deficient in minerals, especially Ca and phosphorous (P) [4]. IN high producing crossbred dairy cows, there is remarkable daily drainage of Ca through milk. Adequate mineral supplementation needs to be devised to replenish the daily loss. There should be care to transient periods of cows for their mineral needs [5].

Calcium and P supplements should be nutritional devices to fortify the normal feeds and fodders to meet the mineral needs of dairy cows at specific levels of productivity [6]. They are essentially required for dairy animals in larger amounts than other minerals. So that, these minerals should be fed at a maximum limit to optimize feed intake and milk production.

In Ethiopia, in most dairy farms especially of stallholders dairy cows do not obtained consistent mineral supplementation as the majority of dairy farms depend on conventional feeds, which may not satisfy mineral needs for optimum growth calves, milk yield and reproduction of cows [7]. Furthermore, dairy farmers lack knowledge of mineral supplementation in ration formulations, leading to impaired dairy animal production, productivity, increased disease incidence, and reproductive problems [8]. There was no previous study on dairy feed mineral supplementation in the study area. Therefore, the present work was undertaken to determine the effect di-calcium phosphate (DCP) supplementation to dairy feed on feed intake, digestibility, and calcium balance, for optimum milk production.

## Materials and methods

### Description of the area

The feeding trial was on-station trial conducted at dairy farm of Addis Ababa University (AAU), College of Veterinary Medicine and Agriculture (CVMA) located in Bishoftu town. Bishoftu town is located 45 km along South East of Addis Ababa, the capital of Ethiopia at 9˚N latitude and 40˚E longitude and 1850 m.a.s.l. The annual rainfall is 866 mm of which 84% is in the long rainy season from June to September. The annual average temperature ranges from 12.3˚C to 27.7˚C, with an overall average of 18.7˚C [9].

### Experimental animals

A total of eight F1 Boran X Fresian cows with 106.0 ± 5.0 (mean ± SD) days in milk, with one to four parities, with an initial body weight of 469.0 ± 15.8 kg (mean ± SD), and with milk yield of 10.0 ± 1.1 l/head/d were selected to be included in this studies. The cows were individually stall-fed in a well-ventilated barn with a concrete floor. All the cows were weighed and drenched with broad-spectrum anti-helminths (Albendazole 2500 mg, please add countries of origin and batch numbers) and checked against subclinical mastitis using California mastitis prior to the start of the experiment.

### Experimental design

Before formulation of experimental feed and creating experimental design, laboratory analysis was made on mineral and chemical compositions of existing feed and experimental feeds were formulated based on the result of mineral and chemical compositions of laboratory analysis (Table 1). The feeds used constituted grass hay as a basal diet, and concentrate consists of noug seed (Guizotia abyssinica) cake, wheat bran, and common salt.

During the base line study different dairy feeds and blood samples were collected, and analyzed samples were identified as sufficient or deficient minerals in Ca, P, Zn and Cu (Tables 2 and 3). The feeds were collected from two districts and corresponding urban and periurban

**Table 1. Chemical and mineral composition of experimental feeds (%DM).**

| Parameters | Hay | NSC | WB | NSC+WB(1:2) |
|---|---|---|---|---|
| DM | 90.50 | 93.70 | 87.07 | 91.30 |
| CP | 6.15 | 32.00 | 15.30 | 27.80 |
| OM | 87.90 | 87.97 | 92.57 | 90.23 |
| Ash | 12.10 | 12.03 | 7.43 | 9.77 |
| NDF | 73.80 | 44.15 | 45.54 | 37.61 |
| ADF | 37.55 | 30.93 | 29.64 | 23.54 |
| ADL | 10.55 | 6.38 | 7.25 | 7.23 |
| Ca% | 0.74 | 0.32 | 1.30 | 0.42 |
| P% | 0.15 | 0.41 | 0.89 | 0.65 |

DM = dry matter; CP = crude protein; OM = organic matter; NDF = Neutral detergent fibre; ADF = Acid detergent fibre, ADL = Acid detergent lignin; NSC = noug seedcake; WB = wheat bran.

**Table 2. Summary of seasonal variation along with production system (%) mineral status from analyzes dairy feeds with relation to their critical level.**

| | | | Districts N = 140 | | | | | | | | |
|---|---|---|---|---|---|---|---|---|---|---|---|
| | | | | Adaa | | | | | Adama | | |
| | | | Urban | | Peri urban | | Urban | | Peri urban | | |
| Mineral | Season | Suff. | Def. | Suff. | Def. | Suff. | Def. | Suff. | Def. | C.* level |
| Ca | dry | 73.37 | 26.67 | 70.13 | 29.87 | 68.33 | 31.67 | 80.33 | 19.67 | < 0.30 |
| | wet | 88.33 | 11.67 | 78.33 | 21.67 | 77.33 | 22.67 | 89.33 | 10.67 | |
| P | dry | 88.68 | 11.32 | 92.66 | 7.34 | 85.55 | 14.45 | 90.33 | 9.67 | < 0.25 |
| | wet | 86.77 | 13.23 | 89.31 | 10.69 | 83.70 | 16.30 | 88.70 | 11.30 | |
| Cu | dry | 62.11 | 37.89 | 52.44 | 47.56 | 73.53 | 26.47 | 53.66 | 46.34 | < 0.8 |
| | wet | 72.92 | 27.08 | 60.65 | 39.35 | 62.61 | 37.39 | 62.55 | 37.45 | |
| Zn | dry | 54.4 | 45.60 | 62.66 | 37.34 | 36.55 | 63.45 | 72.37 | 27.63 | < 0.30 |
| | wet | 46.77 | 53.23 | 53.66 | 46.34 | 64.63 | 35.37 | 68.79 | 31.21 | |

Def. = deficient; suff. = sufficient.

**Table 3. Summary of physiological status along with production system in terms of percentage (%) mineral status of blood from analyzes dairy cows with in relation to their critical level.**

| | Districts | | | | | | | | | |
|---|---|---|---|---|---|---|---|---|---|---|
| Mineral | | | Adaa Adama | | | | | | | |
| | | Urban Peri urban Urban Peri urban | | | | | | | | Critical level* |
| | Phys. State | Suff. | Def. | Suff. | Def. | Suff. | Def. | Suff. | Def. | |
| Ca | Lactating | 76.66 | 23.34 | 70.68 | 29.32 | 76.66 | 23.34 | 81.46 | 18.54 | <0.08g/l |
| | Dry cow | 92.88 | 7.12 | 91.68 | 8.32 | 88.87 | 11.13 | 93.66 | 6.34 | |
| P | Lactating | 50.77 | 49.23 | 67.67 | 32.33 | 70.63 | 29.37 | 73.88 | 26.12 | <0.04g/l |
| | Dry cow | 83.86 | 16.14 | 100 | 0 | 82.57 | 17.43 | 87.56 | 12.44 | |
| Cu | Lactating | 20.46 | 79.54 | 45.38 | 54.62 | 20.77 | 79.23 | 50.65 | 49.35 | <0.65ppm |
| | Dry cow | 42.38 | 57.62 | 32.13 | 67.87 | 32.38 | 67.62 | 40.61 | 59.39 | |

Source: Asian Journal of Dairy and Food Research 2020; Critical level* (Ca<0.08g/l = 8mg/dl, P<0.04g/l = 4mg/dl, P<Cu<0.65ppm, Fe<1.0ppm, Zn<0.6ppm), Co<0.1ppm; adapted from [12] cu = copper; Fe = Iron; Zn = zinc; co = cobalt; phys, state = physiological state; Def. = deficient; suff. = sufficient.

production system in wet and dry season. Following this identification, feeding trial were subjected for supplementation with that of critical deficient mineral (Ca and P) for lactating dairy animals in the form of DCP.

In general experimental treatment feeds consisted of dairy concentrate mix (noug seed cake and wheat bran) served as a base mix in the control diet supplemented with increasing amounts of DCP. Hence, during the feeding trial, the supplements were Ca and P in the form of DCP used as a supplementing 24% Ca and 18% P/25Kg, which was a 0.6% requirement for lactating cows [10]. To meet the requirements, the experimental ration which included wheat bran (72%), noug seed cake (26%), DCP (0.6%) and salt (1.4%). The total ration roughly provides each cow daily with 9.4 MJ/kg DM, 12.3% CP to support 10 liter milk/day with 3.6% butterfat content [10, 11].

Change-over experimental design was employed based on their parities (one to four). Four treatments assigned to the same animal, but in different periods. Each experimental animal is measured more than once, and each measurement corresponds to a different treatment. The total experimental period was consisted of four periods; the length of each period was 30 days (15days for adaptation and 15 days for data measurement) throughout 120 days. The cows were randomly assigned to one of the four dietary treatments, used daily dairy concentrate mix at 0.5 kg $L^{-1}$ of milk produce plus *ad libitum* grass hay (Control = T0), the other treatments were T0 + 0.3% DCP (T1), T0 + 0.6% DCP (T2), and T0+ 0.9% DCP (T3). DCP supplementation was measured daily and mixed with the concentrate feeds every day and required based on the above treatment accordingly.

All groups of experimental animals were fed the corresponding feeds and calcium and phosphorous in the form of DCP divided into two equal portions offered at 0500 and 1700 hours. Grass hay was fed *ad-libitum* (adjusted at 10% daily refusal based). Daily feed offer and refusals were recorded per cow per period to determine the DM, nutrient intake, NDF, ADF, Ca & P composition. Moreover, fecal and urine output were analyzed for DM, CP, NDF, ADF, Ca & P. Individual cows were hand-milked twice a day at 0500 and 1700 hours and milk yield was recorded. The grass hay and concentrate feeds used for feeding weighed daily using sensitive in the morning and evening. Every morning, before feeding, the feed refusals of the previous day were collected and weighed using a similar sensitive balance. The feed refusals of individual experimental animal were collected (300g) in airtight polythene bags, labeled and stored for feed mineral and chemical analyses. The daily dry matter intake for each cow was determined by taking the difference between the amount of feed offered, and the amount of feed refused. Daily CP intake by lactating cows was estimated by calculating the CP in feeds offered by the difference that in the refusals.

### Apparent dry matter feed and nutrient digestibility

Apparent digestibility trial was determined on the same experimental cows used for the feeding trial. The total faecal collection procedure was used for a period of 5 consecutive days at the end of each period (days 26th to 30 th). To minimize faecal contamination, assistant and farm personnel were assigned to scoop the faeces into plastic buckets while the animals defecate. Individual cow's faece was weighed every morning before 0800 hours and before feeds were provided to the animals. However, to avoid possible contamination, faecal samples for laboratory analysis were obtained directly from the rectum at (0800, 1400, 2000 and 0200 hours). Faecal samples from each cow were composited, 1% of the total collection was placed in a polyethylene bag, and stored at -20°C until laboratory analysis. At the end of the collection period, Pooled samples of feed offer and refusals and faecal output were analyzed for DM, CP, NDF, and ADF. Total ration apparent DM and nutrient digestibility were calculated using the

following formula [13].

$$DM/\text{Nutrient digestibility } (\%) = \frac{DM/\text{ Nutrient intake} - DM/\text{Nutrient in feces}}{DM/\text{Nutrient intake}} \times 100$$

## Sources and methods of data collection

**Blood samples collection.** About 10 ml of blood was collected form jugular vein and 2ml serum was harvested and kept at -20˚C until digestion [14].

**Milk samples collection.** During morning and evening milking 100 ml aliquot milk samples were collected in sterile plastic containers for two consecutive days at the beginning (15[th] and 16[th] day) and two consecutive days at the end of the experimental period (29[th] and 30[th] day). The samples were kept at -20˚C untill laboratory analysis for milk fat, protein, lactose, and total solids. Digestion was done at Debreziet Agricultural Research Center for and milk composition analysis was done at Ethiopian meat and dairy industry development institute (EMDIDI). Milk composition analysis was determined following standard methods. Samples were thawed immediately before analysis, mixed thoroughly using vortex mixer and analyzed [15].

**Urine samples collection.** The urine was collected following the procedure of [16]. Urine was collected for 5 days during the last week of the feeding trial (26[th] - 30[th] days) following the procedure of Composites were made from collected urine samples, and 10% sub-sample was prepared and stored following the procedure [16].

**Laboratory analysis.** Representative feed samples were taken and subjected to oven drying (55˚C for 72 h), ground to 1 mm sample size using a Wiley mill, and stored in plastic bags. All samples were analyzed for DM, ash, CP, using the procedure of [15]. Moreover, NDF, ADF, were determined by the procedures of [17]. A two-stage [18] in vitro digestibility technique was employed to analyze and calculate the digestible organic matter in the DM of the samples. Estimated metabolizable energy (EME) was estimated from the in vitro digestible organic matter in the dry matter (DOMD) as $EME\ (MJ/kg) = 0.16 \times DOMD$ [19]. Individual milk composite samples from the morning and evening sample were thawed before analysis, mixed thoroughly using vortex mixer, and analyzed for total solids, protein, lactose, and fat (milk lacto Scan Analyzer, 2016) at Ethiopia meat and dairy development industry (EMDIDI) [20].

**Determination of calcium and phosphorous.** Serum, urine, and milk samples were digested as per the procedure described by [21]. 3 ml of serum was mixed with 3ml of concentrated HNO3 in the tube. The mixtures of sample with concentrated HNO3 were kept overnight at room temperature and digested by heat at a temperature of 70–80˚C until the sample volume was about 1 ml. Then, 3 ml of the double acid mixture (3 part concentrated HNO3 and 1 part 70% $HClO_4$) was added, and low heat digestion continued until the digested samples became clear and emitted white fumes. Similarly, corresponding 10 ml blanks were also prepared following the procedure [22].

All the feed, milk, urine, fecal, and serum samples were analyzed using (AAS) (Perkin-Elmer 300 AAS) at Holota Agricultural Research center (HARC) nutrition lab. Phosphorus concentration in serum samples were determined with photoelectric colorimeter. Instrument was set at zero with blank at DZARC and/or HARC in Ethiopia. The word "critical level" (CL) is used in this study is to express as a mineral concentration in feedstuffs and dairy cows [12]. Ca and P balance was calculated based on the formula [23].

$$Mineral\ (Ca \& P) balance = Ca \& P\ Intake - Ca \& P\ Outgo\ (Milk + Feces + Urine).$$

### Ethics statement

This study was carried out in accordance with the recommendations of Adis Ababa University Animal research ethics. The protocol was approved by the Animal research Review Committee on the Ethics of Animal Experiments of Adis Ababa University (Protocol VM/ERC/07/11/018). All procedures and conditions stipulated in the research are respected and any deviation or changes be reported to the committee.

### Statistical analysis

The analysis of variance (ANOVA) was conducted using the general linear model (GLM) procedure of [24] for windows. Analysis of Variance (ANOVA) model statement used to investigate the effects feeds, blood serum, fecal, urine, milk yield and compositions, DM and nutrient intakes and digestibility. The model used to estimate the variance component was Latin square ANOVA procedure. The mean minerals concentrations of samples were compared statistically using SAS's General Liner Model (GLM) procedure. Mean separation for mineral element concentrations were compared using Tukey's test and significance level were considered at P<0.05. The following statistical models were used:

Model used for the feeding trials (i.e., $4 \times 4$ double change-over design set as Latin Squares):

$$Yij(k)m = \mu + SQm + ROW(SQ)im + COL(SQ)jm + \tau(k) + \varepsilon ij(k)m$$

Yij(k)m = observation/response variables (milk yield, milk chemical compositions, feed DM and nutrient intakes and apparent digestibility for ij(k)m, $\mu$ = the overall mean, SQm = the effect of square m, ROW(SQ)im = the effect of parity i within square m, COL(SQ) jm = the effect of period j (feeding days) within square m, $\tau(k)$ = the effect of a specific treatment k, and $\varepsilon ij(k)m$ = random error.

## Result and discussion

### Effect of dicalcium phosphate on dry matter intake and nutrient intake

The present study showed that DM intake (DMI) increases as the level of DCP supplementation increases in crossbred lactating cows (Table 4). Total DMI was significantly different

**Table 4. Dry matter and nutrient intake (kg/d), and body weight change (kg) of cows fed with different levels of dicalcium phosphate to concentrate mix.**

| Parameter | Supplementation of DCP at different level of % DM/d | | | | | |
|---|---|---|---|---|---|---|
| | 0 | 0.3 | 0.6 | 0.9 | SEM | P-value |
| Hay | 6.47[c] | 7.38[b] | 9.44[a] | 9.43[ab] | 0.22 | 0.0229 |
| Concentrate | 4.99[c] | 5.43[b] | 5.15[ab] | 6.25[a] | 0.19 | 0.0149 |
| Total DMI | 11.46[c] | 12.81[b] | 14.59[ab] | 15.68[a] | 0.40 | 0.0248 |
| T. Nutrient intake | | | | | | |
| CP | 2.1[d] | 2.12[c] | 2.18[b] | 2.21[a] | 0.07 | 0.0561 |
| NDF | 7.03[c] | 7.81[b] | 8.42a | 8.37[ab] | 0.21 | 0.0323 |
| ADF | 3.10[c] | 3.34[b] | 3.79[ab] | 3.80[a] | 0.11 | 0.0134 |
| EME(MJ/kg DM) | 11.25 | 11.60 | 11.93 | 12.33 | 0.24 | 0.0114 |
| Average BWkg$^{-1}$ | 469[c] | 476[b] | 480[ab] | 485[a] | 8.44 | 0.210 |
| DMI/100 kg BW | 2.44[c] | 2.69[b] | 3.03[ab] | 3.23[a] | 0.20 | 0.012 |

Means without superscripts are non-significantly different (P>0.05); DMI = Dry matter intak; CP = crude protein; OM = organic matter NDF = Neutral detergent fibre; ADF = Acid detergent fibre; EME = Estimated metabolizable energy; SEM = Standard error of mean; T1 = grasshay+concentrate+0.3% DCP; T2, grass hay + concentrate 0.6% DCP; T3, hay+concentrate+0.9% DCP; DCP18%;* average Initial body weight of experimental cows was 469.0 ± 15.80kg (mean ± SD); T = total.

**Table 5. Nutrient digestibility (% DM) of lactating cows fed with concentrate mix diets supplemented with dicalcium phosphate.**

| Nutrient digestibility | Supplementation of DCP at different level in % DM/d of DCP at different level %DM/d | | | | | |
|---|---|---|---|---|---|---|
| | **0** | **0.3** | **0.6** | **0.9** | **SEM** | **P-value** |
| DM | 70.54[c] | 71.87[b] | 77.40[ab] | 78.97[a] | 1.04 | 0.0147 |
| CP | 77.80[d] | 78.82[c] | 81.31[b] | 83.11[a] | 1.29 | 0.0215 |
| NDF | 43.31[c] | 45.60[b] | 48.11[ab] | 48.40[a] | 1.87 | 0.0280 |
| ADF | 28.52[d] | 29.35[c] | 30.23[b] | 34.93[a] | 1.76 | 0.0118 |

Means with superscripts are significantly different(P<0.05); DM = Drymatter; CP = crude protein; NDF = Neutral detergent fiber; ADF = Acid detergent fiber; SEM = Standard error of the man.T1 = grass hay+concentrate+0.3%DCP; T2 = grasshay+concentrate 0.6% DCP; T3 = grass hay+concentrate+0.9% DCP.

(p<0.05) among the treatment groups. The mean value for total DMI and nutrient intake of experimental cows in $T_0$, $T_1$, $T_2$, and $T_3$ was 11.46, 12.81, 14.59 and 15.68 kg DM per day for cows received DCP at the level of 0.3, 0.6 and 0.9% DM per day, respectively. DMI was influenced by DCP supplementation, which was significantly different (P < 0.05) compared between treatment and control groups. The current result was in agreement with the report of [25], which indicated that mineral deficiency reduced feed intake and might contain some minerals like P deficiency, resulting in loss of appetite. Likewise, mineral deficiency may result in poor feed utilization, reduced milk yield, lowered disease resistance, increased milk fever incidence, reduced growth rate, osteoporosis, and osteomalacia [16]. Importantly, the calculated DMI per 100 kg animal body weight was 2.44, 2.69, 3.03, and 3.23% for $T_0$, $T_1$, $T_2$, and $T_3$, respectively. This might be due to feeding of Ca has been suggested to improve the performance of dairy cows [26]. Nonetheless, there was a pronounced trend for Ca and P supplementation which there was no significant effect on nutrient digestibility. Interestingly, contrary to the current result [27] indicated that feeding different sources of Ca and P (DCP, rock phosphate and superphosphate) in the diet of calves was not affecting digestibility of organic nutrients (OM, ADF and CP).

The effects of DCP on DM nutrient digestibility of lactating crossbreed dairy cows are indicated in Table 5. Apparent DM digestibility increased with increasing the level of DCP supplementation, which was significantly different (p< 0.05) among the treatment groups. There was evidence that [21] suggested that mineral supplementation improves nutrient digestibility of feeds, which improve animal performance; this might be animals can't perform their genetic potential if their mineral needs are not fulfilled even, their protein energy needs 100% satisfied. Moreover, Ca supplementation affects the digestibility of CP. However, feeding Ca in the excess may reduce the digestibility of protein and energy [16].

## Milk production and composition

This research indicated that supplementation of DCP affected daily milk yield and compositions in Table 6. There were significant differences (P<0.05) in milk yield and milk composition except total solid. The calculated percentage of the average final milk yield by the difference in relation to the average initial milk yield was (0.90, 18.51, 19.50, 19.83%) for lactating cows received DCP at the level of 0, 0.3, 0.6, and 0.9% DM/day of DCP, respectively. Therefore, milk fat, protein and lactose composition were significantly different (P< 0.05) in supplemented groups. The result is supported by [28] indicated that supplementation of DCP to lactating cow had a significant effect (P<0.05) on milk composition and quantity. Likewise supplementation of DCP to lactating cows significantly affected milk fat, milk quantity, and quality of protein compared to the non-supplemented group [29].

**Table 6. Average milk yield (l/d) and milk compositions (%) fed concentrate mix supplemented with dicalcium phosphate.**

| Parameters | Supplementation of DCP at different level in % DM/d | | | | | |
|---|---|---|---|---|---|---|
| | **0** | **0.3** | **0.6** | **0.9** | **SE** | **p-value** |
| Initial milk yield* | 9.95[c] | 10.75[b] | 10.77[ab] | 11.75[a] | 0.35 | 0.0001 |
| Av. Daily milk yield | 10.04[c] | 12.74[b] | 12.87[ab] | 14.08[a] | 0.47 | 0.0161 |
| Difference | 0.09[c] | 1.99[b] | 2.10[ab] | 2.33[a] | 0.01 | 0.0182 |
| Milk composition | | | | | | |
| Fat | 3.62[d] | 4.15[c] | 4.36[b] | 4.73[a] | 0.15 | 0.0096 |
| Protein | 2.85[c] | 3.48[b] | 3.72[ab] | 4.15[a] | 0.18 | 0.0127 |
| Lactose | 3.60[b] | 4.23[ab] | 4.30[a] | 4.30[a] | 0.18 | 0.0211 |
| Total solids | 12.10 | 11.50 | 11.60 | 12.57 | 0.71 | 0.7078 |

[a-d] Means with different superscripts within row are significantly different (P<0.05); T0 = control; T1 = grasshay+concentrate+0.3% DCP; T2, grasshay +concentrate0.6%DCP; T3, hay+concentrate+0.9% DCP; DCP18% initial milk yield = average milk yield taken during adaptation period.

The more daily milk and better milk composition produced from cows received 0.9% DCP, it may be justified by supplementation of Ca and P enhance the milk production and fat content in milk. This was attained by the better utilization of feed DM, which improves the rumen's buffering capacity, which improves the rumen's buffering capacity due to the sufficient presence of Ca, which enhances the rumen's buffering capacity. Moreover, improve an increase in fat is due to increasing Ca in the diet providing more buffering capacity in the gastrointestinal tract, which improves the production of the high ruminal molar percentage of acetic and butyric acids due to the fermentation of fiber.

Moreover, inadequate or lack of mineral supply is the major technical problem that results in low total milk output, reduced milk yield per cow, this might be due to Ca and P deficiency which leads to reduced appetite and milk yield production [30].

Results of serum Ca, and P concentrations of lactating cows are given in Table 7. The serum Ca concentration was significantly different (P<0.05) between the supplemented groups. The low Ca serum concentration in higher DCP supplemented group might be due to dietary Ca may reduce the Ca mobilization from bones for metabolism, resulting in lower Ca level in blood serum [29]. Generally high degree of regulation of calcium homeostasis in the cow maintains plasma calcium at an adequate level in most situations, and as long as the regulatory systems are functioning, plasma calcium level is maintained independent of dietary calcium level [23].

Lactation tends to lower the level of blood Ca as a result of the transfer of blood Ca to the milk [16].

The effect of supplementation of DCP in all treatment groups were significantly different (P<0.05) for total Ca intake, Ca in milk, Ca in faeces, total Ca excretion and balance (Table 8). The net absorption of minerals are affected by the secretion into the gastrointestinal tract, which may be considerably higher than the uptake in the intestine; hence, Ca and P absorption

**Table 7. Effect of dicalcium phosphate supplementation in % DM/d on blood serum Ca and P concentrations in lactating cows.**

| Blood serum | 0 | 0.3 | 0.6 | 0.9 | SEM | p-value | CL(g/l) |
|---|---|---|---|---|---|---|---|
| Ca(g/l) | 0.09[d] | 0.35[a] | 0.22[c] | 0.16[b] | 0.01 | 0.017 | <0.08 |
| P(g/l) | 0.06 | 0.06 | 0.06 | 0.05 | 0.00 | 0.306 | <0.04 |

[ab] Means with different super scripts within row are significantly different (P<0.05); T0 = control; T1 = grasshay+concentrate+0.3%DCP; T2, grasshay+concentrate0.6% DCP; T3, hay+concentrate+0.9%DCP; CL = criticallevel, Ca = <0.08g/l; P = <0.04g/l [12]; SEM = standard error of mean.

**Table 8. Mean ±SE daily intake and Ca and P balances in different groups of dairy cows(g/head/d).**

| Parameters | Supplementation of DCP at different level in % DM/d | | | | | |
|---|---|---|---|---|---|---|
| | 0 | 0.3 | 0.6 | 0.9 | SEM | P-value |
| Hay DMI (Kg/d) | 6.47[c] | 7.38[b] | 9.44[a] | 9.43[a] | 0.22 | 0.0229 |
| Conc. DMI (Kg/d) | 4.99[c] | 5.43[ab] | 5.15[b] | 6.25[a] | 0.19 | 0.0149 |
| Total DMI (Kg/d) | 11.46[c] | 12.81[b] | 14.59[ab] | 15.68[a] | 0.40 | 0.0248 |
| Total Ca intake (g/d) | 64.76[d] | 85.29[c] | 107.81[ba]b | 120.51[a] | 2.23 | <0.0001 |
| Ca in feces (g/d) | 61.89[c] | 72.46[b] | 73.41[b] | 77.77[a] | 1.86 | 0.0165 |
| Ca in urine (g/d) | 2.28 | 2.58 | 2.67 | 2.74 | 0.12 | 0.5239 |
| Ca in milk (g/d) | 9.0[c] | 11.43[bc] | 11.61[b] | 13.0[a] | 0.42 | 0.9332 |
| T. Ca outgo (g/d) | 73.17[c] | 84.47[bc] | 85.95[b] | 93.51[a] | 2.01 | 0.037 |
| Ca balance (g/d) | -8.41[c] | 1.15[bc] | 20.12[ab] | 27.0[a] | 0.70 | <0.0001 |
| Total P intake (g/d) | 35.84[d] | 52.49[c] | 59.98[b] | 73.14[a] | 1.03 | <0.0001 |
| P in feces (g/d) | 31.76[d] | 41.59[bc] | 43.83[b] | 47.12[a] | 2.49 | 0.026 |
| P in urine (g/d) | 1.51[d] | 2.20[bc] | 2.35[b] | 2.71[a] | 0.22 | 0.019 |
| P in milk (g/d) | 1.65d | 3.68[bc] | 3.82[b] | 3.96[a] | 0.31 | 0.018 |
| T. P outgo (g/d) | 34.92d | 47.47[c] | 50.00[b] | 53.79[a] | 1.21 | 0.0139 |
| P balance (g/d) | 0.92d | 05.02[c] | 09.98[b] | 19.35[a] | 1.50 | <0.0001 |

[ad]Means with different superscripts in the same row significantly different; NS = nosignificant; T0, = grasshay+concentrate; T1 = grasshay+concentrate+0.3%DCP; T2, grasshay+concentrate0.6%DCP; T3, hay+concentrate+0.9%DCP; DCP = Dicalciumphosphate; T0 = 0, T1 = 0.3%; T2 = 0.6%; T3 = 0.9% DCP18% T = total; Ca and P balance [31].

can be influenced by diet [23]. Similar findings to the present study, indicated that the extent of excretion of Ca depends on the Ca content in the diet and the requirement of lactating cows [23]. Likewise [32], reported that low faecal excretion of Ca in ruminants fed small amounts of feed with low intakes of Ca; this might be the excretion of Ca depending on the calcium content in the diet and the requirement of the animal. Contrary to the current finding [23] indicated that fed Ca from different sources with similar levels neither fecal nor urinary excretion of Ca varied due to the source of Ca in the diet, in which urine output of Ca has generally been shown independent of Ca in the diet.

DCP supplementation significantly affects (P<0.05) P balances, P total excretion, P in faeces, and total P intake between control and supplemented groups. In agreement with the report of [23] indicated that routes of P excretion are faeces and milk; this may be due to mineral excretion both through urinary and faecal routes may correlate with either the mineral intake or mineral status of the animal [33]. It was further suggested to provide excess dietary Ca and P in mid-to-late lactation when the animal could absorb the minerals to replace body stores that were mobilized. According to the research result of [16], the amount of P excreted is dramatically affected by dietary P intake. In general, the concentrations of minerals in urine and faces are influenced by dietary and animal factors such as water intake and nutrient digestibility [33].

## Conclusion and recommendation

The present study revealed that dairy production and productivity in the study area were affected by mineral deficiencies and imbalances since supplementations of dairy feed with mineral mixture were not common in the study area. DCP supplementation (at 0.3, 0.6, 0.9% DM/day of DCP) with the concentrate mix significantly improved dry matter, nutrient intake, and milk composition. Therefore, based on present result the authors recommended supplying

0.6% DM/day of DCP for dairy feed would optimum dairy cow production and productivity. However, the economic efficiency of supplementation should be evaluated. Similarly, the current results need to be confirmed using larger number of lactating dairy animals.

## Animal welfare statement

The authors confirm that the ethical policies of the journal, as noted on the journal's author guidelines page. Ethical approval for the collection of blood and milk sample from lactating cows in an ethical manner was obtained from the Research Ethics and Review Committee of the AAU.

## Supporting information

**S1 Data. Feeding trial data.**
(XLSX)

## Acknowledgments

I would like to thank Dr. Matiwos Habte, Ms. Workinesh Seid, Mr. Musefa, Mr. Dagne and Mr. Mohammed for their assistance in feed chemical composition and mineral analysis of this research. I would also like to show my gratitude to all laboratory staff of Debreziet and Holota Agricultural center, staff members of Adama and Adaa district Agricultural extension workers for sharing and assistance their precious thing of wisdom with me during fieldwork.

## Author Contributions

**Conceptualization:** Wondewsen Bekele Wondater.

**Data curation:** Wondewsen Bekele Wondater.

**Formal analysis:** Wondewsen Bekele Wondater.

**Funding acquisition:** Tilaye Demissie Ayanie.

**Investigation:** Wondewsen Bekele Wondater.

**Methodology:** Wondewsen Bekele Wondater.

**Project administration:** Wondewsen Bekele Wondater.

**Resources:** Wondewsen Bekele Wondater.

**Software:** Wondewsen Bekele Wondater.

**Supervision:** Wondewsen Bekele Wondater, Tilaye Demissie Ayanie.

**Validation:** Wondewsen Bekele Wondater, Tilaye Demissie Ayanie.

**Visualization:** Wondewsen Bekele Wondater, Tilaye Demissie Ayanie.

**Writing – original draft:** Wondewsen Bekele Wondater.

**Writing – review & editing:** Wondewsen Bekele Wondater, Tilaye Demissie Ayanie.

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
