## [Decision Letter · Decision Letter 0]

12 Dec 2022

PONE-D-22-28223Effect of Dicalcium Phosphate Supplementation on Feed Intake, Milk Composition, and Mineral Balances in Crossbred Dairy CowsPLOS ONE

Dear Dr. Bekele,

Thank you for submitting your manuscript to PLOS ONE. After careful consideration, we feel that it has merit but does not fully meet PLOS ONE’s publication criteria as it currently stands. Therefore, we invite you to submit a revised version of the manuscript that addresses the points raised during the review process.

We look forward to receiving your revised manuscript.

Kind regards,

Ewa Tomaszewska, DVM Ph.D

Academic Editor

PLOS ONE

Journal Requirements:

 "The authors received funding from Adis Abeba university, AGP-II (Agricultural Growth Program) through Debreziet Agricultural institute and Dilla university"

   "No potential conflicts of interest to declare "

7. Please amend the manuscript submission data (via Edit Submission) to include author Tilaye Demissie.

8. We note you have included a table to which you do not refer in the text of your manuscript. Please ensure that you refer to Table 1 in your text; if accepted, production will need this reference to link the reader to the Table.

Reviewers' comments:

Reviewer's Responses to Questions

**Comments to the Author**

1. Is the manuscript technically sound, and do the data support the conclusions?

Reviewer #1: Yes

Reviewer #2: Partly

2. Has the statistical analysis been performed appropriately and rigorously? 

Reviewer #1: Yes

Reviewer #2: Yes

3. Have the authors made all data underlying the findings in their manuscript fully available?

Reviewer #1: Yes

Reviewer #2: Yes

4. Is the manuscript presented in an intelligible fashion and written in standard English?

Reviewer #1: Yes

Reviewer #2: Yes

5. Review Comments to the Author

Reviewer #1: 1. Is the manuscript technically sound, and do the data support the conclusions?

“Yes”, because all the procedure of data collection and analysis was followed the scientific paper procedure

2. Has the statistical analysis been performed appropriately and rigorously?

“Yes”, because the latine square ANOVA procedure was appropriate way for repeated measurement on a single animal, but it need re-writing because there are repeated words and sentences such as; ANOVA, GLM, SAS. On the model the parity was indicated but nothing was explained in the discussion and conclusion about the effect of parity on the indicated results..

3. The authors made all data underlying the findings in their manuscript fully available?

“Yes”, because all the processed or analysed data underlying the findings described in the manuscript are presented in the Table. Otherwise, the raw data that was used for analysis may be with the author.

4. Is the manuscript presented in an intelligible fashion and written in standard English?

“Yes” but it need revision or re-writing because ; some of the sentences are not clear and palatable for readers

Additional comments and question

Title should be modified as:-

Effect of Di-calcium Phosphate Supplementation on Feed Intake and digestibility, Milk Composition, and Mineral Balances in Crossbred of Holestien and Boran Dairy Cows

Abstract:- should be revised and part of the conclusion and recommendation should be included properly.

Objectives:- escaped from the manuscript; therefore, major and specific objectives should be indicated in the manuscript.

Experimental animals

Number of Animals for each Treatment ( T0, T1, T2 and T3) was not indicated, instead total number (8) was mentioned

Animals are clearly indicated but their parity was not clear and not indicated in the result b/s what was the effect of parity on the result ????

You said initial body weight 469.0 ± 15.8 kg, and what was final body weight??? Why not average body weight ???

Experimental design

Table 1, 2 and 3 are not important b/s they show the composition and or deficiency in the previous study of mineral not the level of supplementation for each treatment, instead referring the total deficiency of those minerals in the single sentence and indicating the source is enough.

Result and discussion:-

Need more discussion with good language consistently supported by the recent findings or reference

If you say significant (P<0.05) you have to show the highest and the lowest value of your finding. Becauase I have seen general explanation during your discussion

Conclusion

During conclusion some of the figure or number from the result appear or repeated. it is not correct and it should be concluded only based on the objective of the study in words.

Reviewer #2: Endale etal., 2015 ..Missed from reference part

Birla et.al.,2017—lack of consistency

Section 2.1. last paragraph …Lemma et. al , 2016 -- lack of consistency

The total experimental period was consisted of four periods; the length of each period was 30 days (15days for adaptation and 15 days for data measurement) throughout 120 days…..page 5….. There is contradiction with abstract part… DCP was supplemented for 120 days of feeding trial period classified into four periods, and each consisting of 30 days (16 days of adaptation and 14 days of data collection) and at different levels (0; 0.3; 0.6; 0.9 % DM/day of DCP) for T0, T1, T2, T3, respectively…….so which one is correct …..

(Taylor et al 2007)--------------Lack of Consistency –part 2.5.3.

(Kolmer et al 1951)….part 2.5.5… lack of Consistency

(Dias et al., 2012)….part ..3.1. Lack of Consistency

(Taylor et al., 2009). …part..3.1. Lack of Consistency

Generally, the abstract, conclusion, and recommendation are poorly stated, therefore, the author should be rewritten strongly

6. PLOS authors have the option to publish the peer review history of their article (what does this mean?). If published, this will include your full peer review and any attached files.

Reviewer #1: **Yes: **Tsegaye Eshetu Sime

Reviewer #2: No

---

## [Author Response · Author response to Decision Letter 0]

22 Feb 2023

Reviewer 1.First of all we would like to thank and appreciate the reviewer for their valuable comments and suggestions it will address very well and would significantly improve the quality of our manuscript.

We have accepted most of the suggestions forwarded by the Reviewer and corrected the manuscript accordingly. Where we had a different idea to the reviewers' comments we have kindly tried to explain our point.

Specific responses for specific issues raised by the Reviewers are given below (in blue).

Comments from the editors and reviewers:

1. Reviewer 1.

Most of the issue which raised by the first reviewer was comments about the modification of title, number of animals in each treatment, about the parity of animals, repeated words and sentences such as; Table 1, 2 and 3 are not important b/s they show the composition and or deficiency in the previous study of mineral not the level of supplementation for each treatment, instead referring the total deficiency of those minerals in the single sentence and indicating the source is enough. ANOVA, GLM, SAS 

We have accepted and change comments in in the title, Abstract, title, methodology result and discussion part and conclusion 

A. About title we modified title:” Supplementing Dairy Feed by Dicalcium Phosphate and Effect on Dry Matter Intake, Digestibility, Milk Composition, and Blood Mineral Balances in Crossbred Dairy Cows “

B. About abstract, methodology, result and discussion as well as conclusion and recommendations we accepted the comment and we go through and modified, corrected which are found in manuscript of response to reviewers. 

However, one of the comment raised by the first reviewer’s was about experimental animals: we were a total of eight mid- lactating cows 106.0 ± 5.0 day (mean ± SD) lactation stage, with one to four parities of cross breed of Holstein Friesian X F1 Boran cows were selected with an average initial body weight of 469.0 ± 15.8 kg (mean ± SD). In this case the number of cows in treatment was one (1) since the design was a 4 × 4 double “change-over design” set as Latin Squares in which individual animals can get each treatment. This all indicated in methodology except some modifications.

2. Table 1, 2, and 3 are my Ph.D result which was published in Asian Journal of Dairy and Food Research 2020. This work is my original work that helps the base line information to carry out feeding trial on those deficient minerals which were identified as a deficient, which dominantly affect the milk yield of lactating cows. Therefore, my previous result specially identification of deficient minerals which affect our dairy cows were identified from on farm lactating dairy feeds, dairy blood(serum) and finally this makes to made feeding trial (mineral) to what level supplementation give optimum milk production in our context. That’s why I used this data as a preliminary information as a base for my feeding trial.

3. Another comment is about analysis: SAS is software, and the result is analyzed through analysis of variance (ANOVA), mean minerals concentrations of samples were compared with General Liner Model (GLM) procedure that why I used however, I made some rephrasing.

NB: The rest are comments, we accept and incorporate all.

Reveiwer 2. We have incorporated all of your suggestions into my revisions they were helpful. Thank you 

The second reviewer has more focused on references: 

The total experimental period was consisted of four periods; the length of each period was 30 days (15days for adaptation and 15 days for data measurement) throughout 120 days…..page 5….. There is contradiction with abstract part… DCP was supplemented for 120 days of feeding trial period classified into four periods, and each consisting of 30 days (16 days of adaptation and 14 days of data collection) and at different levels (0; 0.3; 0.6; 0.9 % DM/day of DCP) for T0, T1, T2, T3, respectively.

We accepted as a comment because there was different number in adaptation and data measurement the exact is 16 days for adaptation and 14 days for measurement period 

Except Endale et al., 2015 & Lemma et. al , 2016, the rest sources which are mentioned by second reviewer are available in the document. We have accepted and change

NB.Points made by the Editor thank you for your help here below we incorporate some of the responses for comments raised 

The case of proof reading and the sponsorship

We have accepted and made a change specially some redundant words, absence and presence as well as improper utilization of punctuations and rephrasing some phrases in result and discussion part

The question have raised about the funding source, when I performed this research I was working my Ph.D in Adis Abeba University almost all budget from Adis Abeba university my Dilla University through the thematic project of Dr Tilaye Demissie who is included in publication (Advisor). However, the AGP-II (Agricultural Growth Program) through Debreziet Agricultural research allowed for us the dairy cow farm to perform our feeding trial simply because the Debrezeit Agricultural research need the output of the research and we did that as recommendation.

---

## [Editor Report · Decision Letter 1]

27 Feb 2023

Supplementing Dairy Feed by Dicalcium Phosphate and Effect on Dry Matter Intake, Digestibility, Milk Composition, and Blood Mineral Balances in Crossbred Dairy Cows

PONE-D-22-28223R1

Dear Dr. Wondewsen Bekele Bekele,

We’re pleased to inform you that your manuscript has been judged scientifically suitable for publication and will be formally accepted for publication once it meets all outstanding technical requirements.

Kind regards,

Ewa Tomaszewska, DVM Ph.D

Academic Editor

PLOS ONE
---

## [Editor Report · Acceptance letter]

8 Mar 2023

PONE-D-22-28223R1 

Supplementing Dairy Feed by Dicalcium Phosphate and Effect on Dry Matter Intake, Digestibility, Milk Composition, and Blood Mineral Balances in Crossbred Dairy Cows 

Dear Dr. Wondater:

I'm pleased to inform you that your manuscript has been deemed suitable for publication in PLOS ONE. Congratulations! Your manuscript is now with our production department. 

Kind regards, 

on behalf of

Professor Ewa Tomaszewska 

Academic Editor

PLOS ONE